# Steatotic Livers Are More Susceptible to Ischemia Reperfusion Damage after Transplantation and Show Increased γδ T Cell Infiltration

**DOI:** 10.3390/ijms22042036

**Published:** 2021-02-18

**Authors:** Elke Eggenhofer, Anja Groell, Henrik Junger, Amoon Kasi, Alexander Kroemer, Edward K. Geissler, Hans J. Schlitt, Marcus N. Scherer

**Affiliations:** 1Department of Surgery, University Hospital Regensburg, University of Regensburg, 93053 Regensburg, Germany; anja.groell@t-online.de (A.G.); Henrik.junger@ukr.de (H.J.); amoon.kasi@stud.uni-regensburg.de (A.K.); geissler.edward@ukr.de (E.K.G.); Hans.schlitt@ukr.de (H.J.S.); Marcus.scherer@ukr.de (M.N.S.); 2MedStar Georgetown Transplant Institute, MedStar Georgetown University Hospital, Washington, DC 20007, USA; Alexander.H.Kroemer@gunet.georgetown.edu; 3Division of Personalized Tumor Therapy, Fraunhofer Institute for Experimental Medicine and Toxicology, 93053 Regensburg, Germany

**Keywords:** liver transplantation, ischemia/reperfusion injury, marginal organs, T cells

## Abstract

Liver transplantation (LTx) is often the only possible therapy for many end-stage liver diseases, but successful long-term transplant outcomes are limited by multiple factors, including ischemia reperfusion injury (IRI). This situation is aggravated by a shortage of transplantable organs, thus encouraging the use of inferior quality organs. Here, we have investigated early hepatic IRI in a retrospective, exploratory, monocentric case-control study considering organ marginality. We analyzed standard LTx biopsies from 46 patients taken at the end of cold organ preparation and two hours after reperfusion, and we showed that early IRI was present after two hours in 63% of cases. Looking at our data in general, in accordance with Eurotransplant criteria, a marginal transplant was allocated at our institution in about 54% of cases. We found that patients with a marginal-organ LTx showing evidence of IRI had a significantly worse one-year survival rate (51% vs. 75%). As we saw in our study cohort, the marginality of these livers was almost entirely due to steatosis. In contrast, survival rates in patients receiving a non-marginal transplant were not influenced by the presence or absence of IRI. Poorer outcomes in marginal organs prompted us to examine pre- and post-reperfusion biopsies, and it was revealed that transplants with IRI demonstrated significantly greater T cell infiltration. Molecular analyses showed that higher mRNA expression levels of CXCL-1, CD3 and TCRγ locus genes were found in IRI livers. We therefore conclude that the marginality of an organ, namely steatosis, exacerbates early IRI by enhancing effector immune cell infiltration. Preemptive strategies targeting immune pathways could increase the safety of using marginal organs for LTx.

## 1. Introduction

In liver transplantation (LTx), organ exposure to ischemia and reperfusion during the transplantation process often leads to poor graft function; moreover, it may cause inflammatory activation in the intestine, thus leading to multi-organ failure [1,2]. Unfortunately, a lack of transplantable human livers has led to the usage of “marginal” organs at high-risk for severe ischemia reperfusion injury (IRI) [3].

The need for donor organs has grown due to the increasing incidence of organ failure, while the supply of optimal deceased donors remains insufficient to meet the high demand. There are many approaches to expand the donor pool to meet the prevailing organ shortage. For decades, attempts have been made to meet the steadily growing demand by splitting one donor organ into two [4], by increasing living donor LTx [5], and by domino transplants [6]. Due to the shortage of organs, there is also growing usage of organs from donors after circulatory death [3]; however, in Germany, unlike in some neighboring countries, organ removal after cardiac death is prohibited. Notably, organs from donors after circulatory death are suboptimal for transplantation due to prolonged warm ischemia periods before cold preservation [3]. It must also be considered that the brain death of a donor leads to elevated levels of proinflammatory cytokines such as IL-6, IL-10, TNF-α, TGF-β, and MIP-1α and thus to an increased IRI [7]. Nevertheless, the use of marginal organs is an increasingly considered option for expanding the donor pool and reducing waiting list mortality [8].

In the past, factors contributing to donor organ function loss and marginality have been investigated, e.g., “marginal organs” carry a higher risk of severe IRI, resulting in primary non-function or early poor transplant function [3]. Extended donor criteria have been developed to control the use of marginal organs for transplantation [3]. For example, in LTx, there are specific expanded criteria for donation (ECD) that Eurotransplant uses to determine the marginality of an organ [9]. However, this ECD classification has no consequence for allocation within Eurotransplant [10] and rather represents an exploratory component of organ quality. Indeed, there is no clearly implemented definition of an ECD [10]. In view of this situation, the conditioning and optimal use of marginal organs becomes increasingly important for the further utilization of such organs in the future.

It is known that reperfusion damage results from direct and indirect cytotoxic mechanisms [11]. A deficiency of the antioxidant glutathione is associated with reperfusion damage [12]. At the cellular level, ex vivo cold IRI is dominated by sinus endothelial cell damage and the destruction of the microcirculation [13], with matrix metalloproteinases playing an important role in sinus endothelial cell damage [14]. In contrast, warm hepatocellular damage in situ is mainly mediated by copper cell-dependent cytotoxic molecules [15]. Despite the fact that ischemia and reperfusion normally occur in a sterile environment, there is an activation of the innate and acquired immune system [16], with the main involved cells consisting of Kupffer cells, dendritic cells, neutrophilic granulocytes, T cells, NK cells, and NKT cells [11]. In our own work, we have shown that unconventional CD27^−^γδTCR^+^ and CD4^−^CD8^−^ double-negative T cells are the most important IL-17-expressing effector cells in hepatic IRI. For this work, we used a mouse model of partial warm hepatic ischemia [17].

In our current investigation, we postulate that the pathophysiological processes involved in the development of IRI have a negative impact on the transplant outcome, especially if the recipient has received a marginal organ. Based on this assumption and the results of our animal model, a retrospective, exploratory, and monocentric case-control study was performed to investigate the patho-mechanistic and immunological factors of early IRI in human orthotopic LTx. The cases represent patients with histopathologically confirmed IRI or the liver transplants of control patients that did not show IRI. In these cases, we studied liver function and outcomes over time, and we analyzed standard biopsies taken after cold preparation and two hours after organ reperfusion for immune cell infiltration and gene expression.

## 2. Results

### 2.1. Recipient Characteristics

Transplant function is markedly influenced by the given conditions in a recipient, so some recipient characteristics like underlying diseases and allocation criteria are presented in detail. Written informed consent was obtained from all LTx patients. We examined 46 patients and took two liver biopsies, one before reperfusion (pre-reperfusion) and one after reperfusion (post-reperfusion). The biopsies were then pathologically examined and divided into patients who showed early signs of IRI (IRI group) and those who showed no signs of IRI (noIRI group). An analysis of the basic characteristics revealed that the average age was higher in the group of patients with histopathologically confirmed IRI and male patients developed an IRI more frequently, although there was no significant difference (Table 1). With regard to the constitution of the recipients, a high body mass index (BMI) correlated very weakly with the IRI (Table 1). Furthermore, the duration of the ischemic periods was compared. Cold, warm, and total ischemia time was not significantly different between the two groups (Table 1). Additionally, lab-MELD, exceptional-MELD, and high-urgency listings did not correlate with early pathological signs of IRI (data not shown).

Looking at renal function and subsequent IRI development, patients dialyzed before LTx were significantly more prone to develop IRI (Table 1). This means that post-transplant transplant function is sometimes dependent on general organ system condition, making the recipient’s profile a contributing factor to successful LTx.

### 2.2. Donor Characteristics

Assuming that certain donor characteristics have an influence on the etiology of hepatic IRI and that the donor liver condition is critical for later posttransplant function, a precise consideration of the donor profile is required with regard to basic characteristics, previous diseases, laboratory evaluations, and marginality criteria. Interestingly, a closer examination of donor sex revealed that transplants from male organ donors were significantly more frequently associated with the development of an IRI, regardless of the recipient’s sex. Gender mismatch of donors and recipients was not significantly different (Table 2). Furthermore, with regard to organ perfusion, parameters relevant to the circulation could have an influence on later organ function. Therefore, we analyzed the donor’s need for the catecholamine noradrenaline and blood transfusions. The use of a higher amount of norepinephrine was found to correlate weakly with a reduced frequency of IRI; there was only a very weak correlation of IRI with blood transfusions (Table 2). Likewise, intensive care requiring the therapeutic use of immunosuppressive corticosteroids revealed only a weak correlation with the presence or absence of IRI (Table 2). Furthermore, perfusion solution type and volumes did not significantly affect the two groups (Table 2).

Furthermore, the donor risk index (DRI) [18] was applied on the basis of a modified evaluation of different variables. However, we detected no significant difference in DRI values when comparing patients with the presence or absence of an IRI (Table 2).

The analysis of the basic characteristics thus showed that liver transplants from male donors are significantly more likely to develop IRI. In addition, a therapeutically higher-dose noradrenaline treatment of the donor was found to be associated with a protective effect against IRI damage. Consequently, our data suggested that the characteristics of the donor do not substantially impact IRI prevalence and organ function after transplantation.

### 2.3. Distribution of Patients

Understandably, transplants with a poor organ quality likely also show poorer organ function after transplantation, and that marginal organs carry a greater risk of severe IRI [3]. For this reason, in addition to the classification of the patient population according to early histopathological IRI signs, the further classification of the transplants of this patient population was also carried out by analyzing different donor and transplant characteristics (Table 1 and Table 2) that, according to Eurotransplant, define the marginality of a transplant [9]; these characteristics are donor age > 65 years, ICU stay with ventilation > 7 days, BMI > 30, steatotic liver > 40%, serum sodium > 165mmol/L, serum glutamic-pyruvic transaminase (GPT) > 105U/L, serum GOT > 90U/L, and serum bilirubin > 3 mg/dL (Table 3). When at least one of these criteria exceeded this defined limit, the corresponding transplant was classified as a marginal organ [9].

Organs from donors with a high BMI and histopathologically proven hepatic steatosis over 40% developed IRI significantly more frequently (Table 3). Similarly, serum sodium was significantly different and higher in the group of donors whose transplants developed an IRI after LTx (Table 3). The other parameters (serum transaminases, age, ICU stay, and bilirubin) showed no difference between the IRI vs. noIRI groups.

In the Eurotransplant region, 50% of liver transplants come from donors with expanded donor criteria [10]. In accordance with this information, marginal liver transplants were allocated in 54% of the cases (*n* = 25) in the patient collective of this study (*n* = 46) (Table 4). Interestingly, 68% (*n* = 17) of the patients receiving a marginal organ developed a histopathologically confirmed IRI (*p* = 0.447). Therefore, these results suggested that the allocation of marginal organs likely contributes to the development of hepatic damage due to ischemia and reperfusion.

We subdivided the patient population into the following groups: group 1 separated the patients according to whether they had histopathological early IRI; group 2 separated the patients according to whether they had a marginal liver; and group 3 divided patients without IRI with a non-marginal transplant, patients without IRI with a marginal transplant, patients with IRI with a non-marginal transplant, and patients with IRI and a marginal transplant (Table 4). The incidence of patients with IRI and marginal transplantation was found to be almost twice as frequent as the frequency of patients without IRI with non-marginal transplantation, which underlined the importance of IRI and marginality in clinical transplantation medicine.

### 2.4. Patient Survival

We compared the survival of patients without IRI with a non-marginal graft, patients without IRI with a marginal graft, patients with IRI with a non-marginal graft, and patients with IRI and a marginal graft (Figure 1). Patients with IRI and a marginal transplant were found to die significantly more frequently and to have a significantly lower probability of survival compared to patients with IRI and a non-marginal transplant.

There was a significant difference between IRI patients with a marginal and IRI patients with a non-marginal transplant within the third group classification (Table 3) in terms of the probability of survival. Similarly, the biggest difference in the one-year survival rate was also between the IRI + M group (51%) and IRI + n-M group (91%).

### 2.5. Clinical Outcome

The preoperative laboratory parameters of the recipients and values over the first seven post-operative days (PODs) were analyzed. In particular, we measured parameters representing liver cell damage, cholestasis, and the synthesis performance of the liver, as well as metabolic end products. We compared patients with the presence or absence of IRI, as well as patients with a marginal organ with regard to the presence or absence of IRI. Interestingly, GPT did not differ statistically when comparing the IRI vs. noIRI patient populations in the preoperative phase, although higher GPT levels tended to be found in the IRI group postoperatively (Figure 2A, left). An analysis of the total bilirubin when comparing IRI and noIRI patients indicated that IRI tended to be associated with a higher bilirubin level, with significant differences already occurring at the first and second PODs (Figure 2A, right).

While the mean GPT value was increased preoperatively in the group of patients with a nonmarginal transplant, significantly higher values were found in the group of patients with marginal transplantation at the first, second, and sixth PODs. An analysis of patients with marginal grafts additionally revealed that patients with IRI had significantly higher liver cell damage at POD three, six, and seven compared to patients without IRI (Figure 2B, left). Patients with a marginal allograft plus additional IRI had significantly higher total bilirubin levels compared to patients who had not developed an IRI on PODs one-to-seven (Figure 2B, right).

After looking at the clinical laboratory data after transplantation, it became clear that IRI and the marginality of organs not only led to liver cell damage (as measured by GPT, GOT, and GLDH) but also to a reduced hepatic synthesis performance (as measured by factor V) and an increase in the metabolic end product bilirubin; these results showed the effects of IRI and marginality in the clinical course. Therefore, given the significantly poorer survival associated with IRI and marginality, it is important to determine possible contributing cellular mechanisms and genetic factors in human LTx.

### 2.6. Cellular Mechanisms

#### 2.6.1. Histological Examination of Infiltrating T cells

We have shown in an animal model that, in connection with 90 min of ischemia and a subsequent 24 h of reperfusion, a considerable infiltration of CD3+ T cells occurs. Our analysis revealed that in addition to CD4-CD8-CD3+ T cells, unconventional NK1.1-CD27-γδTCR+CD3+ T cells were a main source of IL-17A, which is an important cytokine involved in IRI damage [17]. Therefore, based on these results, we immunohistochemically examined pre- and post-reperfusion liver biopsies to quantify CD3+ T cell infiltration in human LTx.

We found significant differences regarding the infiltration of CD3+ cells into portal fields post-reperfusion, indicating that T cells do infiltrate IRI-damaged livers. However, we could not detect significant levels of γδ T cell infiltration, though this was likely due to the low levels of cell numbers in tissue sections (data not shown).

#### 2.6.2. Gene Expression Analysis

Our previous animal model IRI investigations showed that unconventional NK1.1^−^CD27^−^γδTCR^+^CD3^+^-T cells are the most prominent infiltrating cell population driving hepatic IRI, together with CD4^−^CD8^−^CD3^+^-T cells [17]. Based on these data, we decided to use a sensitive and specific gene expression analysis to detect CD3^+^ and γδTCR^+^ cells in human LTx biopsies, as well as the chemokine CXCL-1. In support of the histological results, an association of high CD3 mRNA expression with the IRI transplants was evident (Figure 3B). Interestingly, the biopsies prior to reperfusion already showed considerably higher mRNA expression levels in the IRI group, which could be attributed to ischemic pre-transplant damage. Transplants with IRI showed an increase in CXCL-1 mRNA expression; in comparison to patients without histopathological IRI, the difference before reperfusion was found to be significantly different. To examine this further, we analyzed the γδ T cell loci and found a significant increase in post-reperfusion expression in grafts with IRI compared to grafts without IRI (Figure 3B). Together, these results supported the premise that the immune system plays an important role in human hepatic IRI.

## 3. Discussion

IRI is a central problem of transplantation surgery that is increasingly becoming the subject of intensive research due to the shortage of organs and associated allocation of marginal donor organs. Indeed, marginal organs carry a higher risk of developing an IRI [3]. In our study, we found that IRI patients experience a longer stay on the ICU, which has been reported by others [19], and have a higher rate of primary allograft non-function [20]. Our current study cohort showed an IRI occurrence rate of 63% by biopsy pathology after reperfusion. A similar study by Gaffey et al. detected organ preservation damage in 70% in post-LTx biopsies [21]. Kakizoe et al. compared liver biopsies before transplantation and after re-perfusion by light microscopic analyses and also found a high prevalence of IRI in post-reperfusion biopsies [22]. However, the lab-MELD, exceptional MELD, and high-urgency listings of the recipients were not correlated to early pathological signs of IRI (data not shown). While various contributing factors to IRI are being researched, there is increasing evidence that CD3+ cells play a central role in mediating early IRI. However, the phenotype of T cells driving the cell-mediated immune response and the clinical relevance of these cells remain unknown for human IRI. Notably, T cells are suppressed by standard immunosuppressive drugs, including calcineurin inhibitors, and are neutralized with anti-thymocyte globulins; how these treatments affect transplant-related IRI is not clear. In the current study, we evaluated aspects of immune infiltration, as well as other basic donor and recipient factors that could be contributing to the IRI problem.

The analysis of donor sex showed that transplants from male donors, regardless of recipient sex, are more likely to develop an IRI, although gender mismatch is not a factor. In contrast, a clinical study of 436 liver transplanted patients did not find donor–gender-associated differences in primary graft non-function and delayed graft failure due to ischemic injury [23]. One reason for this difference may be that their study did not primarily analyze IRI, rather focusing on transplant failure as defined by patient death or need for re-transplantation [23]. Additional studies will be necessary to confirm our findings.

To quantify the risk of transplant failure, Feng et al. developed a DRI by using a combination of several donor variables, like donor age over 40 years, organs from cardiac-dead donors and split/partial LTx, African-American race, small body size, cause of death by cerebrovascular event, and other causes of brain death. There is a significant correlation between the DRI and transplant outcomes within the Eurotransplant region [10]. Since transplant failure may be related to IRI, we analyzed DRI in our patient collective. However, we did not find significant differences in the DRI level when considering IRI versus non-IRI donor organs. Therefore, DRI does not appear to be a predictor for the development of IRI.

We have shown that transplants from donors with high glucose levels are more likely to have an IRI, which could indicate a diabetic metabolic state but could also be an effect of hormones, drugs, stress situations, or brain damage. However, this result should be interpreted with caution, since according to the instructions of Eurotransplant, the donor’s glucose level should be kept within certain limits, e.g., by means of insulin and glucose application [9]. Donors with organs developing IRI were also found to have significantly higher serum sodium levels compared to those without IRI. High donor serum sodium levels are known to be a potential risk factor for transplant dysfunction [3] and significant early transplant loss in the first three months after LTx [24]. It is postulated that cell swelling and the exacerbation of the reperfusion-mediated damage are responsible [3]. As expected, liver-specific GPT was elevated in patients with IRI and showed slower normalization during the first seven postoperative days compared to patients without IRI. The difference became more substantial when considering patients who developed IRI after receiving a marginal organ. A detailed analysis of serum bilirubin also showed differences comparing patients with or without IRI, particularly at the first two PODs. Considering marginal organs and the presence or absence of IRI, serum bilirubin values differed at all tested PODs. Therefore, simple blood tests could provide hints as to whether IRI will develop in transplanted livers and reflect, at least to some degree, the marginality of the donor organ.

BMI and the fatty degeneration of the liver parenchyma were examined in our study, as these reflect the level of organ steatosis [25]. We were able to show that steatosis above 40% and a high BMI are more frequently associated with IRI development. Similarly, Briceño et al. were able to show in a clinical study that steatosis over 30% had an influence on IRI development; moreover, steatosis over 30% was the main cause of severe IRI in that study [19]. In addition, grafts with severe fatty degeneration more often lead to primary non-function [26,27]. Importantly, we found in our study that patients with a marginal transplant who developed an IRI have substantially poorer outcomes, including early mortality. The one-year survival rates in the group of marginal allograft recipients with or without IRI were 51% and 75%, respectively. Interestingly, we observed that survival in recipients receiving a non-marginal transplant was not appreciably affected by IRI development. Previous studies looking at recipients receiving organs from donors with high BMI indices observed early graft dysfunction in 56% of recipients, and those patients showed shorter graft survival outcomes. Another study showed that donor livers with more than 30% steatosis were associated with a significant decrease in both four-month and two-year recipient survival [28,29]. Together with our results, we conclude that organ marginality, including the influence of high donor BMI, substantially affects LTx and patient outcomes.

One objective of our study was to investigate immune cells that could be involved in IRI. We previously showed in an animal model that hepatic IRI leads to CD3+ T cell infiltration [17]. An analysis of pre- and post-reperfusion biopsies of our current patient collective showed that transplants with IRI had greater CD3+ cell infiltration. Interestingly, Jassem et al. reported higher levels of CD3+ lymphocytes in the livers of deceased donors compared to living donors [30], which was likely related to longer ischemia times with deceased donors. It is also important to note that Jassem et al. observed substantial CD3+ cell infiltration before reperfusion, supporting our hypothesis that CD3+ could play a role in IRI. Indeed, our gene expression analysis showed increased CD3 and CXCL-1 mRNA in the pre-reperfusion biopsy samples. Since we previously showed in an animal model that unconventional NK1.1-CD27-γδTCR+CD3+ cells are important effector cells in IRI [17], we confirmed the presence of γδTCR mRNA in our human post-LTx perfusion IRI-affected tissues. Interestingly, γδ T cells have been discussed in the context of IRI, but these were mainly animal models [31,32,33,34]. It is also intriguing, and consistent with γδ T cell involvement in IRI, that numbers of these cells decrease in the peripheral blood following renal IRI, presumably due to trafficking to the transplanted kidney [35]. We therefore suggest that more in-depth studies are needed to better define the role of T cells in liver IRI.

To the best of our knowledge, this is the first study to identify γδT cells in pre- and post-reperfusion biopsies in LTx. Therefore, this work contributes to understanding the mechanisms related to early IRI, and it will provide a basis for developing specifically targeted therapies. Minimizing early IRI would especially benefit patients receiving marginal organs and thus improve the outcomes after LTx.

## 4. Materials and Methods

### 4.1. Patients and Biopsy Samples

The study was approved by the ethics committee of University Hospital Regensburg (18 Aug 2011, ethical approval code #11-101-0163) and was registered on the University Hospital Regensburg Clinical Trials Register (Z-2012-0048-6). After written informed consent, 46 consecutive patients participated in the study. All data regarding the donor were retrieved from Eurotransplant International Foundation (EIF, Leiden, The Netherlands). The DRI was validated as calculated by Feng et al. [18]. The grade of steatosis of the donor liver and IRI scoring in the recipient were evaluated by the Institute of Pathology, University Medical Center Regensburg. Data about age, gender, BMI, underlying conditions, and survival were retrieved from patient documentation. Data regarding dialysis and MELD were retrieved from EIF. Overall ischemic time was calculated as the sum of cold and warm ischemic time using data from the surgical reports and EIF.

Two biopsies were obtained from each graft—one at the end of back-table preparation and the second at the end of liver transplantation prior to abdominal closure. Biopsy samples were split and placed into formalin and RNAlater solutions (Qiagen, Hilden, Germany).

### 4.2. Blood Samples

During liver transplantation, one systemic arterial blood sample was collected at the end of liver transplantation prior to abdominal closure, and then a sample was collected daily after transplantation (routine blood drawings on the ward). All samples were analyzed in the Department of Clinical Chemistry at the University Hospital Regensburg.

### 4.3. (Immuno)histochemistry (IHC)

The liver biopsies were assessed for the histological features (H&E staining): parenchymal neutrophilic infiltration, portal inflammation, hepatocellular necrosis, and presence of apoptotic cells to determine early hepatic IRI in the Department of Pathology at the University Hospital Regensburg. Liver damage (percent necrosis) was determined morphometrically in paraffin sections (3 μm). Slides were stained with H&E and an antibody against CD3 (Fisher scientific GmbH, Schwerte, Germany) according the routine standards of the Department. The areas photographed with a reversal microscope (Zeiss AxioVision Observer Z1, Carl Zeiss Microscopy GmbH, Jena, Germany) were evaluated with the image processing program ImageJ (NIH, Bethesda, MD, USA), which was used to quantify the positively stained cells in a supported manner. The infiltration of CD3+ cells was investigated in the portal fields. A portal field was used for the quantification if at least 2 lumina and fibrotic parts were visible. All completely visible portal fields of the CD3-stained biopsies were evaluated by calculating the cell density of each portal field on the basis of cell count per area in mm^2^ and then determining the mean value from the individual cell densities. The number of portal fields per liver punch biopsy varied from 1 to 12 depending on the size of the cut surface. For relative quantification, the quotient of the mean value of the liver punch biopsy after reperfusion to the mean value of the liver punch biopsy before reperfusion was determined for each patient.

### 4.4. Real-time PCR (RT-PCR)

cDNA was isolated from human liver samples using the µMACS ONEstep cDNA Kit (Miltenyi) according to the manufacturer’s protocols. cDNA was then amplified by PC with SybrGreen^®^ gene expression Master Mix (Roche, Basel, Switzerland). Predesigned primer sets for GAPDH (QT01192646), CXCL-1 (QT00199752), and CD3 (QT00001477) were used, while primers for γδTCR were designed as follows: forward—ccccaagcccactatttttc; reverse—caagaagacaaaggtatgttccag (Roche, Basel, Switzerland). Within each experimental group, mRNA expression was normalized to GAPDH amplification (ΔCt), and then fold changes relative to noIRI preLTx (ΔΔCt) were determined using 2^−(ΔΔCt)^ [36].

### 4.5. Statistics

All data were continuously collected, updated, documented, and analyzed using the IBM^®^-SPSS^®^-Statistics-20 (IBM Deutschland GmbH, Ehlingen, Germany) statistics software.

All continuous variables are given as mean values with standard deviations. The test for normal distribution was performed using the Shapiro–Wilk test. Two groups with independent, normally distributed, continuous variables were evaluated with the Student’s *t* test on independent samples while taking variance homogeneity into account. Two groups with non-normally distributed continuous variables were analyzed using nonparametric tests such as the Mann–Whitney U test. The comparison of normally distributed, continuous variables of more than two groups was analyzed using ANOVA. Non-normally distributed continuous variables from more than two groups were compared using the Kruskal–Wallis H-test. Correlation analyses of interval-scaled and normally distributed variables were performed with the product-moment correlation according to Pearson. If one of the two variables was ordinally scaled or not normally distributed or if one of the two variables was dichotomous, the rank correlation according to Kendall’s tau was used. If both variables were dichotomous, the correlation was calculated using four-field correlation with distance and similarity measures according to Russel and Rao. The correlation was given with the correlation coefficient *r*.

To determine cumulative patient survival, defined as the time between transplantation and death or end of data collection, Kaplan–Meier curves were generated and checked for statistical significance using the log-rank test.

The box plots each show the median and the 25% and 75% percentiles, with outliers that lie between one and a half and three box lengths outside the box marked with circles and extremes that lie over three box lengths outside the box marked as oval shapes. Differences with *p* < 0.05 were evaluated as significant.

## Figures and Tables

**Figure 1 ijms-22-02036-f001:**
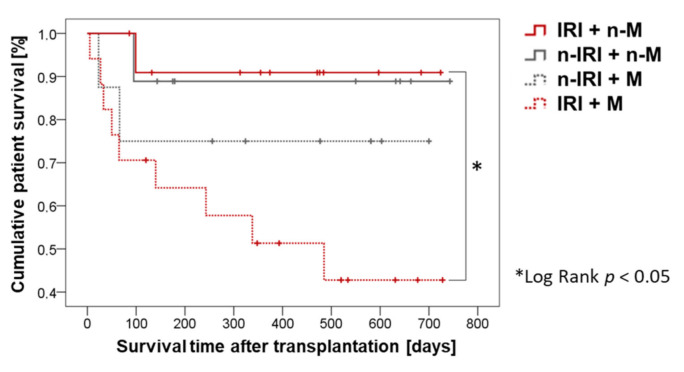
Cumulative patient survival compared to patients with existing or absent histo-pathological IRI and marginal and non-marginal grafts.

**Figure 2 ijms-22-02036-f002:**
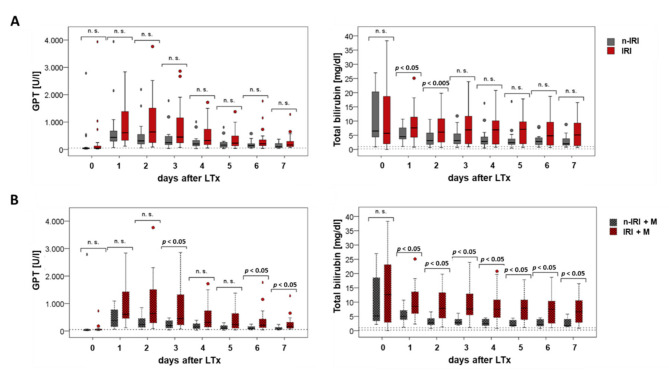
Recipient-GPT and total bilirubin pre-operative (post-operative day (POD) 0) and on PODs 1–7: comparison between patients with an existing or absent IRI (**A**) and between patients with a marginal transplant and an existing or absent IRI (**B**).

**Figure 3 ijms-22-02036-f003:**
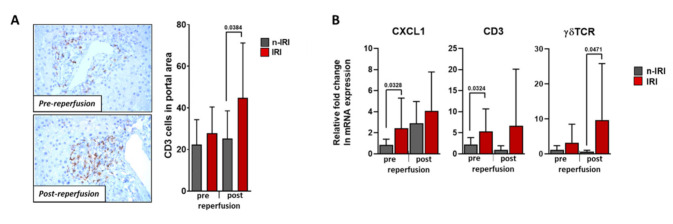
Histological and RT-PCR evaluation of liver allograft biopsies pre- and post-reperfusion. Histological quantification and comparison between patients with an existing or absent IRI (**A**) and RT-PCR analysis of CXCL-1, CD3, and γδTCR (**B**).

**Table 1 ijms-22-02036-t001:** Recipient characteristics: comparison of patients with existing or absent histo-pathological ischemia reperfusion injury (IRI). Number of patients are in round brackets. Mean with range and number of patients is in parentheses. Abbreviations: SD, standard deviation; BMI, body mass index; and LTx, liver transplantation.

	Mean ± SD (*n*)			
	Total	No-IRI Group	IRI Group	*p*
Age (years)	50.11 ± 12.55 (46)	46.47 ± 14.82 (18)	52.24 ± 10.71 (28)	0.214
Male (%)	71.74 (33)	64.71 (11)	75.86 (22)	0.505
Female (%)	28.26 (13)	35.29 (6)	24.14 (7)	0.971
BM (kg/m2)	28.90 ± 8.227 (46)	27.91 ± 9.001 (18)	29.48 ± 7.844 (28)	0.275
Dialysis pre LTx (%)	36.96	17.65	48.28	**0.038**
Cold ischemic time (h)	8.747 ± 2.397 (46)	8.734 ± 3.057 (18)	8.754 ± 1.990 (28)	0.982
Warm ischemic time (h)	0.824 ± 0.177 (46)	0.836 ± 0.186 (18)	0.818 ± 0.175 (28)	0.756
Total ischemic time (h)	9.571 ± 2.397 (46)	9.570 ± 3.035 (18)	9.572 ± 2.008 (28)	0.998

**Table 2 ijms-22-02036-t002:** Donor characteristics: comparison of donor criteria regarding recipients with existing or absent histo-pathological IRI. Number of patients is in round brackets. Mean with range and number of patients is in parentheses. Abbreviations: SD, standard deviation; UW, University of Wisconsin; and DRI, donor risk index. Missing data: DRI data were missing for 2 patients.

	Mean ± SD (*n*)			
	Total	No-IRI Group	IRI Group	*p*
Male (%)	50.00 (23)	29.41 (5)	62.07 (18)	**0.032**
Gender Mismatch, total (%)	43.48 (20)	47.06 (8)	41.38 (12)	0.708
Gender Mismatch, ♂ → ♀ (%)	21.74 (5)	20.00 (1)	22.22 (4)	1.000
Gender Mismatch, ♀ → ♂ (%)	65.22 (15)	58.33 (7)	72.73 (8)	0.667
Noradrenaline (µg/kg/min)	0.218 ± 0.409 (39)	0.306 ± 0.532 (16)	0.156 ± 0.292 (23)	0.055
Blood transfusion (Number of bags)	1.780 ± 4.599 (46)	1.650 ± 4.636 (17)	1.860 ± 4.658 (29)	0.903
Corticoide therapy (%)	45.65 (21)	47.06 (8)	44.83 (13)	0.883
Custodiol^®^-Perfusion solution (%)	89.13 (41)	94.12 (16)	86.21 (25)	0.637
UW-Perfusion solution (%)	10.87 (5)	5.882 (1)	13.79 (4)	0.637
Aortic perfusion volume (L)	12.25 ± 17.96 (46)	13.94 ± 22.25 (17)	11.26 ± 15.26 (29)	0.675
Portal perfusion volume (L)	3.800 ± 1.643 (5)	2.000 (1)	4.250 ± 1.500 (4)	-
DRI	1.400 ± 0.289 (44)	1.388 ± 0.321 (16)	1.407 ± 0.275 (28)	0.833

**Table 3 ijms-22-02036-t003:** Continuous variables of the marginality criteria according to Eurotransplant related to the donor and the transplant. SD, standard deviation; GPT, glutamic-pyruvic transaminase.

	Mean ± SD (*n*)			
	Total	No-IRI Group	IRI Group	*p*
Age (Years)	47.91 ± 15.15 (46)	47.12 ± 13.86 (17)	48.38 ± 16.08 (29)	0.546
ICU Stay (days)	4.337 ± 7.314 (46)	2.459 ± 2.311 (17)	5.437 ± 8.918 (29)	0.122
BMI (kg/m^2^)	25.42 ± 4.484 (46)	24.05 ± 4.386 (17)	26.23 ± 4.414 (29)	**0.045**
Steatosis >40% (%)	19.57 (9)	- (0)	31.03 (9)	**0.017**
Serum-GPT (U/L)	52.54 ± 67.11 (46)	46.00 ± 39.12 (17)	56.38 ± 79.52 (29)	0.750
Serum-GOT (U/L)	66.93 ± 96.79 (45)	78.94 ± 111.0 (17)	59.64 ± 88.46 (28)	0.582
Serum bilirubin (mg/dL)	11.90 ± 7.722 (46)	11.49 ± 8.314 (17)	12.14 ± 7.494 (29)	0.793
Serum sodium (mmol/L)	147.5 ± 6.080 (46)	145.0 ± 5.232 (17)	148.9 ± 6.152 (29)	**0.033**

**Table 4 ijms-22-02036-t004:** Distribution of patients grouped by early post-perfusion injury and marginal organ and the respective combinations: classification of the pre- and post-transplant biopsies of 46 patients according to presence (IRI) or absence (nIRI) of early ischemia reperfusion damage (group 1), as well as receiving a marginal (M) or non-marginal (nM) organ (group 2). Group 3 shows the resulting combinations.

**Group Classification (%)**	**1**		**2**	
	n-IRI	IRI	n-M	M
	39.96 (17)	63.04 (29)	45.65 (21)	54.35 (25)
	3			
	n-IRI and n-M	n-IRI and M	IRI and n-M	IRI and M
	19.57 (9)	17.39 (8)	26.09 (12)	36.96 (17)

## Data Availability

The data presented in this study are available on request from the corresponding author.

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
