# Peer review of "Steatotic Livers Are More Susceptible to Ischemia Reperfusion Damage after Transplantation and Show Increased γδ T Cell Infiltration"

_ijms, 2021, doi:10.3390/ijms22042036_

Round 1
Reviewer 1 Report
I read with interest the manuscript by Elke Eggenhofer et al, its an interesting and well written paper. The authors analysed the incidence of IRI in marginals liver donors and its impact on the overall survival.
I think that the paper could be very useful for the transplant community that work on this issue. The authors must better delineate the definition of IRI, Its not clinical (post reperfusion hynotropic use/ post reperfusion blood pressure etc)? the histological IRI definition must be provide and referenced.
Author Response
Ijms, Author’s reply to the Review Report (Reviewer 1)
Manuscript:
Title Steatotic livers are more susceptible to ischemia reperfusion damage after transplantation and show increased γδ T cell infiltration
Authors Elke Eggenhofer * , Anja Groell , Henrik Junger , Amoon Kasi , Alexander Kroemer , Edward Geissler , Hans Schlitt , Marcus Scherer
Point-by-point Response:
Dear Reviewer 1:
We would like to thank you very much for the very well-founded and positive review.
You are absolutely right in saying that the definition of IRI should have been better defined in our work, as it is a fundamental basis for evaluating the data. The classification and scoring was done by the review of a pathologist at our institute and we have added and detailed the histological review criteria and the IRI definition itself in the methods section as you advised (manuscript line 373-378).
Reviewer 2 Report
I enjoyed your work and found it very helpful in considering ways to prevent IRI as you have demonstrated it is accompanied by activation of inflammatory cells that may possibly be preventable.
I have found that your English is at times stilted and that your points may not be as clearly stated as they could be.
Issues I would like you to address:
1.) The title of your paper suggests that it concerns steatotic livers, yet in the abstract no mention is made of steatotic livers. I suggested correcting that deficiency.
2.) I had been struggling to determine how you define marginal livers. I finally came across this sentence in lines 146-147: "When at least one of these criteria exceeded the defined limit, the corresponding transplant was classified as a marginal organ [9]." I believe you meant to say "liver" instead of "transplant". Were the criteria used these?
Donor age > 65 yrs
ICU stay with ventilation > 7 days; BMI >30;
Steatotic liver > 40%;
Serum Sodium > 165 mmol/l; SGPT > 105 U/l;
SGOT > 90 U/l;
Serum Bilirubin > 3 mg/dl.
If so, I would list them in the body of the paper, as I do not feel the majority of your readers know them.
Editing issues:
- Line 43: You meant circulatory, not circulator.
- Tables 1, 2, & 3: The data listed are said to be means +/- range but in line 397 it is supposed to be standard deviations. Which is it?
- Table 1, there are commas instead of periods for the SD of the first averages of WIT and Total Ischemia time.
- In Table 4's header, I suggest replacing "obtaining" with "receiving".
- In Table 4's body, please add a "M" so that the first patient group in Group 3 is n-IRI + n-M.
- In Table 4's body, align 17.39 (8) under n-IRI + M.
- Delete the stay after ICU in line 256.
- In line 310, change "this" to "that".
- In line 354, replace "conducted" with "calculated".
- Starting in line 168, I suggest rewriting the sentence starting after the colon, "group 1 separates the patients according to whether they had histopathological early IRI; group 2 separates the patients according to whether they had a marginal liver;"
- The graph of patient survival could be better understood if instead of a "s." next to the comparison line you put "p<0.05".
- In that same graph, I would suggest listing in the Legend the group names in the same order that they appear on the graph to make it easier to understand and results. Thus I would list n-IRI + n-M first, then IRI + n-M second, and so forth.
- Line 234, there is no subject for the verb "could".
Data Analysis:
I am confused about your analysis. In the first paragraph about results, you talk at length about correlations and mention r coefficients but you do not show the data. I am not sure whether it is important as the means are not different. If important, I would list it.
Author Response
Ijms, Author’s reply tot he Review Report (Reviewer 2)
Manuscript:
Title Steatotic livers are more susceptible to ischemia reperfusion damage after transplantation and show increased γδ T cell infiltration
Authors Elke Eggenhofer * , Anja Groell , Henrik Junger , Amoon Kasi , Alexander Kroemer , Edward Geissler , Hans Schlitt , Marcus Scherer
Point-by-point Response:
Dear Reviewer 2:
We would like to thank you very much for the very thorough, comprehensive and positive review.
Regarding your criticism of the partly awkwardly worded passages, we have had the entire manuscript proofread (and improved) again by a native speaker and we thank you for bringing this to our attention.
Regarding the points you raised:
- You are absolutely correct in stating that the steatotic livers mentioned in the title did not appear in the abstract. We have changed this (manuscript line 21, 22 and 28).
- Thank you very much for the hint. Yes, of course we meant "livers" and not "transplant". The marginality criteria we used are exactly the ones you listed. We had only cited them as a literature source, but it is really better to list them as well. We have implemented this in the manuscript line 148-150.
All editing issues you found we have improved (editing issues 1-13). Thank you very much for the effort you have put into the review. Our manuscript benefits greatly from it.
Your helpful suggestions about the confusion part in the data analysis (r coefficients) was corrected (manuscript line 95-97 and 123-126).